# Transient ocular hypertension remodels astrocytes through S100B

**Weiran Huang**[1], **Kenji Matsushita**[1]*, **Rumi Kawashima**[1], **Susumu Hara**[1,2,3,4], **Yuichi Yasukura**[1], **Kaito Yamaguchi**[1], **Shinichi Usui**[1], **Koichi Baba**[1,4,5], **Andrew J. Quantock**[6], **Kohji Nishida**[1,3,4]

1 Department of Ophthalmology, Osaka University Graduate School of Medicine, Suita, Osaka, Japan, 2 Transdimensional Life Imaging Division, Institute for Open and Transdisciplinary Research Initiatives, Osaka University, Osaka, Japan, 3 Integrated Frontier Research for Medical Science Division, Institute for Open and Transdisciplinary Research Initiatives, Osaka University, Suita, Japan, 4 Premium Research Institute for Human Metaverse Medicine (WPI-PRIMe), Osaka University, Suita, Japan, 5 Department of Visual Regenerative Medicine, Division of Health Sciences, Osaka University Graduate School of Medicine, Suita, Osaka, Japan, 6 School of Optometry and Vision Sciences, Cardiff University, Wales, United Kingdom

* kenmatsu@ophthal.med.osaka-u.ac.jp

## Abstract

Glaucoma is a series of irreversible and progressive optic nerve degenerations, often accompanied by astrocyte remodeling as the disease progresses, a process that is insufficiently understood. Here, we investigated the morphology of retinal and optic nerve head (ONH) astrocytes under mechanical stress, and explored whether a specific phase is present that precedes astrocyte remodeling. A mouse model of transient ocular hypertension (OHT) and an *in vitro* cell stretch model were established to mimic the pathological conditions of increased intraocular pressure and mechanical stress on cultured cells. Glial fibrillary acidic protein (GFAP), S100B, and actin staining were used to characterize astrocyte morphology and cytoskeleton, with qPCR used to measure mRNA expression. We also silenced S100B expression and conduct RNA sequencing on ONH astrocytes. Astrocytes displayed weaker GFAP intensity (p < 0.0001) in the early-stage OHT mouse model, prior to the onset of hypertrophy, which was accompanied by an increase in GFAP mRNA expression (p < 0.0001) and a decrease in S100B mRNA expression (p < 0.001). *In vitro*-stretched astrocytes tended to contract and had fewer cellular processes and more elongated cell bodies. Downregulation of S100B expression occurred in in both the *in vivo* (p = 0.0001) and *in vitro* (p = 0.0023) models. S100B-silenced ONH astrocytes were similarly characterized by a slender morphology. In the RNA-seq analysis, genes downregulated by more than fivefold were predominantly enriched in terms related to nutrient metabolism, motor proteins and morphogenesis. Meanwhile, genes upregulated by more than fivefold were primarily associated with terms related to histone modification and visual perception. As an early response to mechanical stress, S100B expression is downregulated in astrocytes, which assume a slender morphology, reminiscent of cell "weakening." Silencing intracellular S100B expression induced similar morphology changes and altered the transcriptome. Stress-induced changes were reversible, with evidence of enhanced late-stage reactivation that is likely related to S100B.

**Data availability statement:** All relevant data are within the manuscript and its Supporting information files.

**Funding:** This study was supported by Japan Science and Technology Corporation, Grants-in-Aid for Scientific Research C (No. 24K12782, No. 26462686); Japan Science and Technology Corporation, Grants-in-Aid for Scientific Research A (No. 25253093); Japan Science and Technology Corporation, JST SPRING, Grant Number JPMJSP2138, Next generation Researcher development program; Japan Agency for Medical Research and Development, 23gm1210004.

**Competing interests:** The authors have declared that no competing interests exist.

## Introduction

Glaucoma refers to a group of diseases of multifactorial etiology that are characterized by progressive optic nerve degeneration, and which may cause irreversible blindness. Disease onset is insidious and early detection and diagnosis challenging [1]. Intraocular pressure (IOP) is the most common predisposing and controllable factor [2] and is highly, but not completely, correlated with disease onset and progression [3]. Neuronal and glial cells, peripheral connective tissue, and blood flow are susceptible to damage by elevated IOP compared to IOP in the normal range [2]. Therapy to lower IOP can only slow disease progression and not reverse neurodegeneration [4]; thus, early diagnosis of glaucoma and subsequent neuroprotective therapies are urgently needed.

Astrocyte reactivation can be a consequence of multiple triggers [5]. Reactivated astrocytes have received increasing attention in recent years as promising target cells for interventions to treat glaucoma. Although evidence for astrocyte reactivation derives from patients with glaucoma and experimental animal models of glaucoma [6,7,8], the underlying process remains controversial.

Glial fibrillary acidic protein (GFAP) is an important component of astrocyte intermediate filaments that helps maintain cell shape, which has role in various physiological and biological activities of astrocytes and the regulation of astrocyte–neuron interactions [9]. S100B is a calcium-binding secreted protein, mainly derived from astrocytes, that functions as an intracellular regulator and extracellular signal [10]. It has been widely studied in biological fluids as a biomarker of neurological damage [11,12]. S100B is involved in various cellular processes and regulates cytoskeleton dynamics, $Ca^{2+}$ homeostasis, protein phosphorylation, as well as cell growth and differentiation [13]. At low doses or physiological levels, S100B promotes neuronal growth and survival [14,15] through the Ras-MEK-ERK1/2-NF-κB pathway [10], while, at high doses, it leads to neuronal apoptosis [16] owing to the overproduction of reactive oxygen species and the activation of caspase-3 [10]. Elevated levels of S100B may trigger cytotoxic damage through the s100-RAGE proinflammatory axis [17]. Here, we used a transient mouse model of ocular hypertension (OHT) and an *in vitro* cell stretching model to simulate the influence of glaucoma on astrocytes. The cells were then studied to characterize the early response of astrocytes to mechanical stress in order to gain knowledge that may provide us with new therapeutic strategies and drug targets.

## Materials and methods

### Animals

C57BL/6J male mice and pregnant C57BL/6J female mice, 14–18 weeks of age, were purchased from Charles River Laboratories Japan Inc. (Yokohama, Japan). The male mice were used for the OHT experiments, and the pups of the female mice were used to isolate astrocytes. All procedures conformed to the Association for Research in Vision and Ophthalmology Statement for the Use of Animals in Ophthalmic and Vision Research and were approved by Osaka University's Institutional Animal Care and Use Committee (approval ID 26-034-009). Animals were housed under suitable conditions, with free access to standard rodent chow and boiled tap water. Analgesics and anesthetics were administered, and animal welfare was closely monitored throughout the experiments. All experiments were performed in accordance with the Animal Experiment guidelines of Osaka University.

### OHT induced using ocular viscosurgical devices (OVD)

The mice were anesthetized with a mixture of 0.3 mg/kg medetomidine hydrochloride (Domitor; Meiji Seika Pharma Co., Ltd., Tokyo, Japan), 4 mg/kg midazolam (Dormicum; Astellas

Pharma Inc., Tokyo, Japan) and 5 mg/kg butorphanol (Vetorphale; Meiji Seika Pharma Co., Ltd.), after which tropicamide (Santen Pharmaceutical Co., Ltd) was applied to each treated eye to dilate the pupil to facilitate surgery. After pupil dilatation was achieved, an OVD connected to a 35G needle was inserted obliquely through the cornea into the anterior chamber (to avoid injuring the iris), and approximately 2 μL of the viscoelastic material was injected to completely fill the chamber. Additional injections were performed (up to two) through the original injection site to ensure that the IOP remained elevated for at least 3–4 hours. Contralateral eyes were left untreated. Prior to the OVD injection, IOP was recorded as the mean of three sequential measurements taken with a Tonolab tonometer (iCare Tonovet Co., Ltd., Finland). Following surgery, IOP was measured hourly for 12 hours and again at 24 hours. Mice that developed signs of inflammation were excluded from the study.

## Astrocyte isolation and culture

We referenced the anatomical location described in [18] for harvesting the optic nerve heads (ONHs), which included the intraocular portion of the nerve and the retrobulbar nonmyelinated portion, dissecting the ONHs from 10–15 pups to a distance of 0.6–0.8 mm behind the globe. We then followed the isolation steps outlined in [19] with some adjustments. Briefly, ONHs from C57BL/6 mice pups were isolated, triturated, and placed in 1 mL DMEM (Catalog#:11885084; Gibco) solution containing 20 U/mL papain, 1 mM L cysteine, 0.5 mM EDTA, and 0.005% DNase (Worthington) at 37°C for 30 min. The tissue suspension was then centrifuged at $300 \times g$ for 5 min and resuspended in DMEM containing 2 mg/mL ovomucoid (Worthington) and 0.005% DNase (Worthington). The cells were subsequently washed twice with DMEM and centrifuged at $1,000 \times g$ for 10 min prior to seeding and cultivation in DMEM low glucose with 10% FBS (Catalog#:26140079; Gibco) and 100 μg/mL penicillin/streptomycin (Catalog#:15140122; Gibco) in 5% $CO_2$ at 37°C. The astrocytes were used when their density reached $3–5 \times 10^4$ cells/cm$^2$ regardless of the number of seeded cells. The medium was changed every 2–3 days. All experiments were conducted using primary cells without passaging. For cell identification, we used morphological criteria and immunostaining with GFAP (rabbit; 1:200; G3893; Sigma-Aldrich) and S100B (guinea pig; 1:200; 287004, SYSY). To assess cell viability, we employed the Cell Count Reagent SF assay. Cells were cultured in a solution consisting of Cell Count Reagent SF and culture medium at a ratio of 1:10. After two hours of incubation, the absorbance was measured at 450 nm.

## Immunofluorescence

The eyeballs and primary-culture astrocytes were collected. The eyes were fixed in 4% PFA for 2 hours and the cells for 10 min, both at room temperature. Three paired retinas were dehydrated in graded methanol solutions to obtain retinal flat mounts, while frozen sections were cut from the other eyes. In both procedures, 5% normal donkey serum was used as the blocking agent (1 hour at room temperature). The retinal flat mounts were incubated overnight at 4°C with primary (anti-GFAP [rabbit; 1:200; G3893; Sigma-Aldrich], anti-S100B [guinea pig; 1:200; 287004; SYSY] and anti-Brn3a [rabbit; 1:200; ab245230; abcam]) and secondary antibodies (species-specific secondary fluorescent antibodies goat anti-rabbit Alexa Fluor 488 [1:200; Invitrogen] and goat anti-guinea pig 647 [1:200; ab150187; abcam]). Counterstaining was with 4',6-diamidino-2-phenylindole (DAPI) for 1 hour. The sections were blocked, after which the primary antibody was applied overnight at 4°C, followed by the secondary antibody for 1 hour at room temperature and counterstaining with DAPI for 10 min. The catalog numbers for the secondary antibodies are as follows: Alexa Fluor 488 goat anti-rabbit (#A11008), Alexa Fluor 594 goat anti-rabbit (#A11012), Alexa Fluor 657 goat anti-guinea pig (ab150187).

A similar procedure was followed for the cells, but, after completion of the antibody staining protocol, 100 nM of acti-stain 488 phalloidin (Catalog #PHDG1; Cytoskeleton) was applied for 30 min to stain F-actin. Three-dimensional Z-series datasets were acquired using a confocal laser scanning microscope (Olympus FV3000). For each retina, 12 rectangular areas (three from each quadrant) were imaged. The Z-projection function in ImageJ was used to convert the Z-series images into 2D projected images for further analysis.

### Actin imaging and small interfering RNA (siRNA) transfection

A CellLight Actin-GFP BacMam 2.0 system (Invitrogen) was used to visualize actin. To silence gene expression, the cells were transfected with Lipofectamine RNAimax (Invitrogen), Negative Control duplexes (12935112; Invitrogen), and S100B siRNA Oligo Stealth (MSS276930; Invitrogen). All the transfection steps were performed according to the manufacturers' instructions. siRNA-transfected cells in six chambers were subjected to quantitative polymerase chain reaction (qPCR), while others were immunostained 24 h post transfection. The stretching experiments were conducted 3 days after actin-GFP transfection.

### Real-time qPCR

mRNA was extracted from retinas using ISOGEN II (NIPPON GENE CO., LTD., Japan) as described in the manufacturer's protocol, and cDNA was synthesized using PrimeScript RT Master Mix (Catalog#: RR036B; TAKARA). Twenty-fold diluted raw cDNA was used as template for real-time fluorescent qPCR using the THUNDERBIRD® SYBR® qPCR Mix (Catalog#:QPS-201; TOYOBO CO., LTD., Osaka, Japan) and the Applied Biosystems 7,500 FAST real-time PCR system. The reaction mix was pre-denatured at 95°C for 20 s, followed by 45 cycles at 95°C for 3 s and 60°C for 30 s. Finally, a melt curve was generated by heating the PCR products to confirm the absence of primer dimers. Each primer sample was run in triplicate. The primer sequences were as follows: GAPDH, forward 5′- GCTCATGACCACAGTCCATGC -3′, reverse 5′-ATGCCAGTGAGCTTCCCGTTC-3′; GFAP, forward 5′- CAGGAGTACCAGGATCTACTC-3′, reverse 5′-GTACAGGAATGGTGATGCGGTT-3′; and S100B, forward 5′- CCTCATTGAT GTCTTCCACCAG-3′, reverse 5′-CTCCTTGATTTCCTCCAGGAAG-3′; Brn3a, forward 5′- AGCACAAGTACCCGTCGCTG-3′, reverse 5′-CTGGCGAAGAGGTTGCTCTG-3′. Relative gene expression levels were normalized with GAPDH and fold-change values were determined using the $2^{-\Delta\Delta Ct}$ formula.

### Retrograde tracing of retinal ganglion cells (RGCs)

RGCs were retrogradely labeled with FluoroGold (FG) 7 days before the mice were euthanized by applying a piece of sterile gelatin sponge that had been pre-soaked with 5% Fluoro-Gold (Fluorochrome, Denver, CO) on the surface of the superior colliculus. This procedure did not result in any obvious side effects. The eyeballs were collected after 7 days and immersion-fixed overnight in 4% formaldehyde prior to cryotomy to generate frozen sections or adherence of retinal flat mounts onto glass slides.

### Mechanical cell stretch stimulation

Astrocytes at a density of $3–5 \times 10^4$ cells/cm² were seeded on a stretch chamber STB-CH-24 (STREX Inc., Osaka, Japan) coated with 10 μg/mL laminin. An STB-150 stretching apparatus, modified to stretch up to 40%, was integrated with an inverted microscope (AxioObserverZ1+ $CO_2$ module; Zeiss), with mechanical stimulation experiments conducted at 37°C and 5% $CO_2$. Mechanical stimulation was followed by a gradual stretch–release cycle of 4% or 20% extension at a frequency of 1 Hz.

### RNA sequencing and analysis

Total RNA of siRNA treated astrocytes was initially extracted using Isogen II, followed by further clean-up with the RNeasy Micro kit (cat. no. 74004, Qiagen). Library preparation was conducted using the TruSeq Stranded mRNA Library Prep Kit (Illumina) following the manufacturer's instructions. Sequencing was performed on an Illumina NovaSeq 6000 sequencer using 101-base single-read mode. Sequenced reads were aligned to the mouse reference genome (GRCm38) with HISAT2 v2.1.0. Fragments per kilobase of exon per million mapped fragments (FPKM) were quantified using Cuffdiff version 2.2.1. The top 2,000 most differentially expressed genes were visualized through k-means clustering analysis in iDEP. Furthermore, genes exhibiting at least a five-fold change in expression underwent Gene Ontology (GO) enrichment analysis via Metascape to identify significantly enriched biological terms.

### Image and statistical analyses

ImageJ was used to measure mean pixel intensity and total area above the fluorescence threshold for GFAP and S100B in the regions of interest. The data are presented as the mean value ± standard deviation (SD), and analyzed using GraphPad Prism 9 (GraphPad, La Jolla, CA, USA). Unpaired t-tests and one-way ANOVA tests were used for statistical comparisons, with $p < 0.05$ considered to indicate statistical significance.

## Results

### Elevated IOP affects RGC survival and the state of astrocytes

OVD injection into the anterior chambers of C57BL/6J mouse eyes leads to development of acute and transient glaucoma via obstruction of the aqueous outflow and a sharp rise in IOP. Over time, the viscoelastic material breaks down and IOP returns to normal (Fig 1A). Control eyes retained a baseline IOP of 8.6 ± 1.1 mmHg. In contrast, IOP in OVD-treated eyes increased to 34.6 ± 8.0 mmHg post-injection, which decreased to 27.6 ± 9.1 mmHg 6 hours after the injection, with a further decrease close to baseline values (12.9 ± 2.9 mmHg) after 1 day. Transient IOP elevation, we note, is not considered sufficient to induce chronic optic nerve injury [6]; the FG retrograde labeling data presented here indicated that FG-positive RGCs progressively decreased in number in the transient OHT eyes, with most cells loosing label after 8 weeks (Fig 1C, moderate level damage at each group). The immunostaining for Brn3a, an RGC marker [20], also revealed that RGCs experienced insults of varying severity, from mild to severe (S1A Fig) after 4 weeks. The area of FG labeling (S1B Fig) in the naïve eyes was 31,699 ± 8,882 μm², and 8,917 ± 5,273 μm² ($p < 0.0001$) in the OHT eyes at week 4, with the Brn3a area (S1C Fig) measuring 24,403 ± 18,569 μm² compared to the naïve value of 58,492 ± 17,342 μm², $p < 0.0001$). The qPCR data pointed to a downregulation of Brn3a expression (S1D Fig), with a significant fold-change of 0.95 ± 0.35 in the naïve and 0.35 ± 0.29 in the OHT eyes ($p = 0.0011$). These data suggest that the death of RGCs, not just axonal dysfunction, occurred in OHT eyes compared to the control group.

In the retinas of untreated mice, the astrocytic processes were parallel to the layer of the ganglion cells and their axons (Fig 1Da), forming an ordered network (Fig 1Ea). The distribution of S100B, on the other hand, showed a dramatic difference in concentration between the soma and processes (Fig 1Ea). Fig 1D–1K show the variable response to the elevated IOP, consistent with a report in the literature [21]. In general, three types of changes were observed: wider GFAP coverage (hypertrophy-1), enhanced GFAP fluorescence intensity (hypertrophy-2), and no noticeable changes in GFAP (non-hypertrophy).

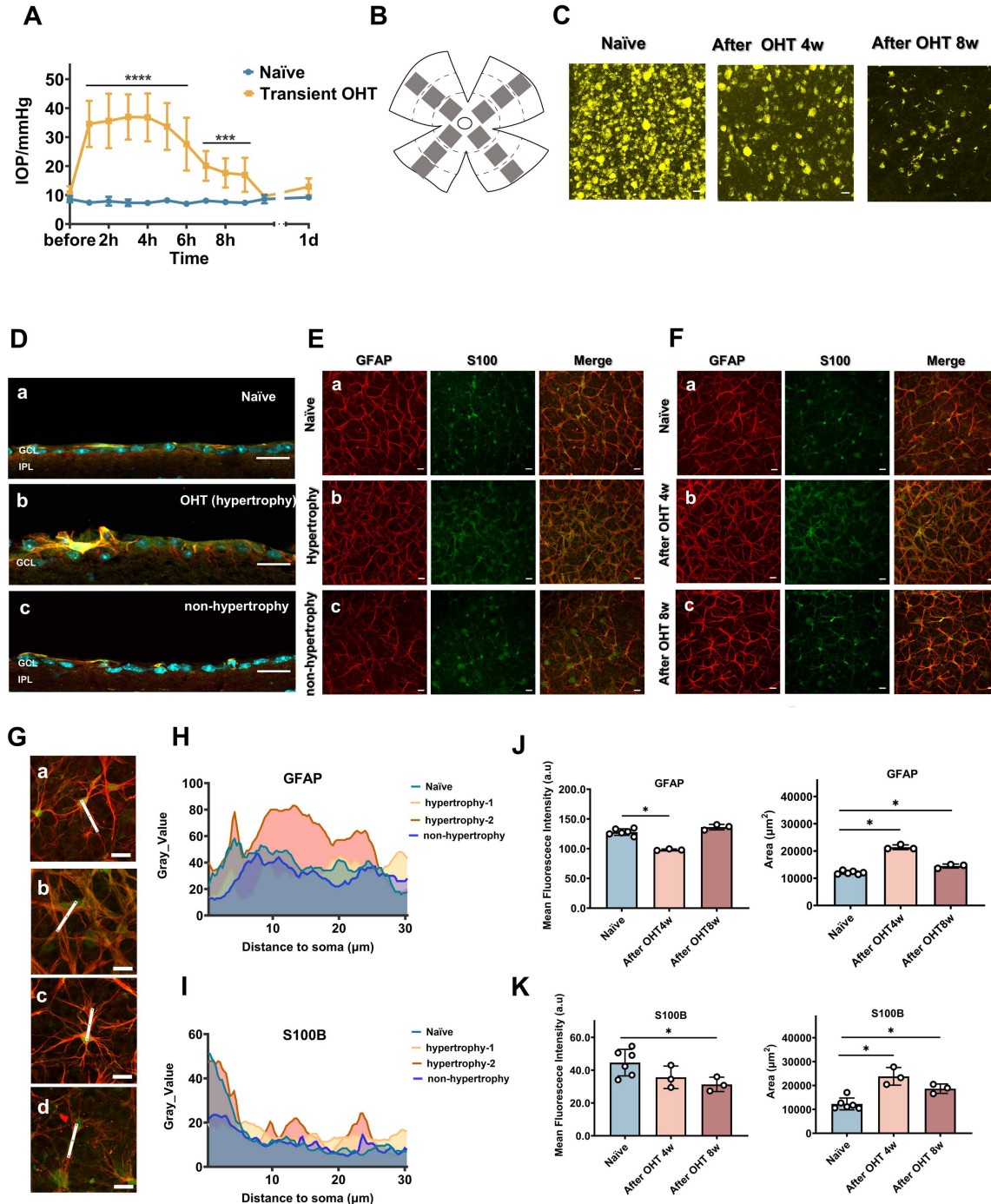

**Fig 1. OVD-induced transient OHT in mice leads to RGC loss and astrocyte reactivity.** (A) OVD injections led to IOP increase, with values presented as means ± SD. IOP decreased 6 hours post-injection and by day 1 had returned to near baseline levels. For the OHT group IOP, n = 110 mice; for the naïve group IOP, n = 6 mice. Within 6 hours, the difference was highly significant with a p-value < 0.0001. Later, at 7–9 hours, the OHT groups continued to show significant differences compared to controls, with a p-value of 0.0004. Unpaired t-test, parametric test, * p < 0.05, ** p < 0.01, *** p < 0.001, and **** p < 0.0001. (B) Schematic diagram of the imaging field distribution in the retinal wholemounts. The gray frames indicate the imaging fields. (C) RGCs were retrogradely labeled with FluoroGold (FG). Short-term IOP elevation can trigger RGC loss after 4 or 8 week. n = 3 mice per group. (D–F) Astrocyte reactivity. Astrocytes were labeled with antibodies against intermediate filaments (GFAP, red) and cytoplasm (S100B, green). GCL nuclei were stained with DAPI (cyan). Da, Ea and Fa: Retina from untreated mice; Db and Eb: Retina from OHT mice with astrocyte hypertrophy after 4 weeks; Dc and Ec: Retina from OHT mice without astrocyte hypertrophy. Fb and Fc: Hypertrophic type astrocytes

at 4 weeks (hypertrophy-1) and 8 weeks (hypertrophy-2). (G) Enlarged image of naïve (a), hypertrophy-1(b), hypertrophy-2 (c) and non-hypertrophy (d) astrocytes. The white line in E represents the measurement line, starting from the cell soma and following the cell process for a distance of 30 μm. Using the "profile plot analysis" in ImageJ, the data (gray values) are shown in panels H and I. (H) Grayscale value comparison of GFAP (white line in G) in astrocytes, showing different patterning. (I) Grayscale value comparison of S100B (white line in G) in astrocytes showing different patterning. (J) Quantification of the fluorescence intensity of GFAP staining and area above the threshold. Individual data points are representative of individual mice. (K) Quantification of the fluorescence intensity of S100B staining and area above the threshold. Individual data points are representative of individual mice. n = 6 mice for the naïve group, n = 3 mice each for OHT post-4 and 8 weeks, with 12 areas imaged in each retina for each group. Unpaired t-test, Mann–Whitney test, * p < 0.05, ** p < 0.01, *** p < 0.001, and **** p < 0.0001. Scale bar: 20 μm.

Thick, vertically growing astrocytic somas and processes were observed in stained sections (Fig 1D). Some of the astrocytes in the transient OHT retinas were widespread and diffusely stained with GFAP, and displayed overlapping processes that extended into each other's domains (Fig 1Eb and Fb). Some astrocytes exhibited strong intermediate filaments and had a complex structure and more "rigid" appearance, in contrast to the diffuse GFAP staining. These cells were relatively distant from each other and promoted the remodeling of astrocyte networks (Fig 1Fc). However, occasionally, especially after 4 weeks, astrocytes did not appear to be significantly hypertrophic (Fig 1Dc and 1Ec), the so-called non-hypertrophic type. In naïve eyes, S100B is mostly concentrated in the soma of astrocytes rather than the processes. However, in some transient OHT eyes, astrocytes lose this high signal of somatic S100B (Fig 1I). The intensity of GFAP staining shown in Fig 1J in the naïve eyes was 128.0 ± 5.0 a.u, and 98.1 ± 1.1 a.u (p = 0.0238) and 136.3 ± 3.8 a.u (p = 0.0952) in the OHT eyes at weeks 4 and 8, respectively. GFAP fluorescence in the naïve eyes extended over an area of 12,054 ± 543 μm², and 21,389 ± 709 μm² (p = 0.0238) and 14,470 ± 607 μm² (p = 0.0238) in the transient OHT eyes at weeks 4 and 8, respectively. The parameters of another specific marker of astrocytes, S100B, were also altered (Fig 1E and 1F), such that the transient OHT retinal astrocytes showed weakened S100B fluorescence intensity but increased area of coverage (Fig 1K). S100B intensity in naïve eyes was 44.6 ± 7.4 a.u, whereas in the transient OHT retinas it was 35.6 ± 5.6 a.u (p = 0.02629) and 31.3 ± 3.6 a.u (p = 0.0476) at 4 and 8 weeks, respectively. The S100B-labeled area in the naïve eyes was 12,265 ± 2,268 μm², and, in the treated eyes, 23,814 ± 3,013 μm² (p = 0.0238) and 18,658 ± 1,606 μm² (p = 0.0476) at 4 and 8 weeks, respectively. Collectively, these data indicate that a transient and dramatic increase in IOP can lead to RGC insult and multiple astrocytic alterations, possibly accompanied by a redistribution and abnormal exocytosis of S100B.

## GFAP/S100B alterations

To investigate the early stage of the remodeling process and determine whether a phase prior to remodeling is evident, we euthanized mice 6 hours post-injection and found that GFAP immunoreactivity was reduced. Indeed, GFAP labeling in the OHT eyes appeared very weak and disorganization was evident (Fig 2A–2C), with the intensity measuring 83.8 ± 8.5 a.u (compared to the naïve value of 128.0 ± 5.0 a.u, p = 0.0012) and the area measuring 11,771 ± 420 μm². As for S100B, the somatic intensity was also reduced, at 34.6 ± 3.3 a.u., compared to the naïve value of 44.6 ± 7.4 a.u. (p = 0.0140), with total coverage measuring 14,405 ± 2,083 μm².

To gain a clearer overview of the dynamic alterations regarding GFAP and S100B in the early and late stages after OHT induction, we plotted all analyzed values of the fluorescence images. The fluorescence intensity of S100B remained low, while that of GFAP recovered over time (Fig 2Da). The expanded GFAP- and S100B-labeled areas indicated astrocyte hypertrophy (Fig 2Db). Although the upregulation of GFAP mRNA expression (Fig 2E)—fold-change

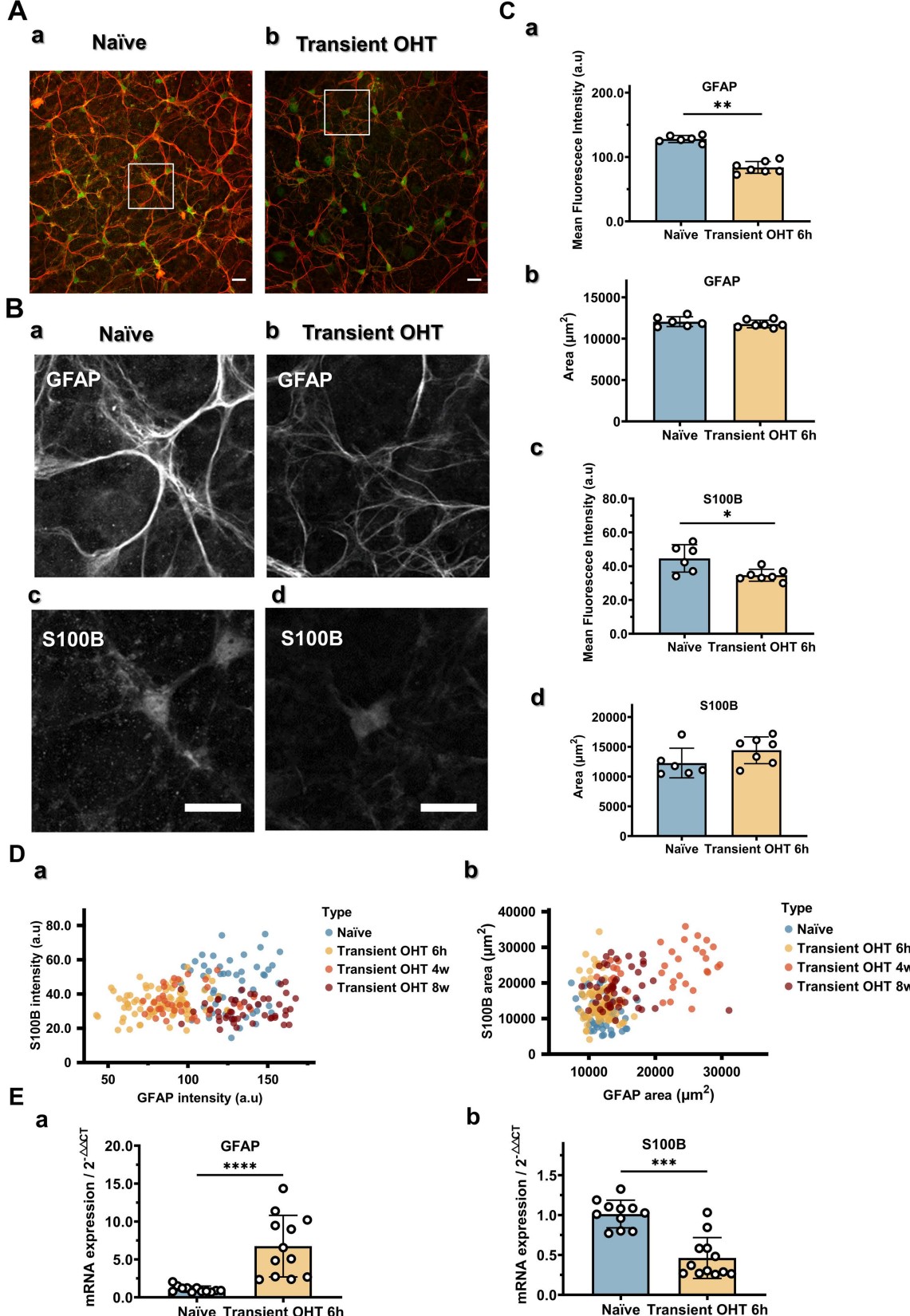

**Fig 2. Cytoskeletal hypotrophy and fluctuation of GFAP and S100B occur during the astrocyte reactive process.** (A) Representative images showing the astrocyte cytoskeleton in an untreated (a) and transient OHT eye (b), stained with GFAP (red) and S100B (green).

Astrocytes in the OHT eyes were thinner and possessed a more disordered network of intermediate filaments than those in untreated eyes. (B) Magnified grayscale images showing the intensity in the areas indicated in A. (a) GFAP in untreated astrocytes. (b) GFAP in transient OHT astrocytes. (c) S100B in untreated astrocytes. (d) S100B in transient OHT astrocytes. (C) Quantification of GFAP (a and b) and S100B (c and d) fluorescence intensity and area above threshold. n = 6 mice for the naïve group, n = 7 mice for OHT post-6 h, with 12 areas imaged in each retina for each group. (D) Fluctuation in GFAP and S100B signals during the reactive process. Individual data points representative of individual filed images are plotted. (E) GFAP (a) and S100B (b) mRNA expression levels measured using qPCR. n = 12 mice each for the naïve and OHT post-6 hour groups. Unpaired t-test, Mann–Whitney test, * p < 0.05, ** p < 0.01, *** p < 0.001, and **** p < 0.0001. Scale bar: 20 µm.

of 1.06 ± 0.39 in the naïve and 6.75 ± 3.89 in the OHT model (p < 0.0001)—was not consistent with diminished fluorescence signal (Fig 2A–2C), the qPCR data pointed to a downregulation of S100B expression (Fig 2E), with a significant fold-change seen—1.01 ± 0.17 in the naïve and 0.46 ± 0.25 in the OHT eyes (p = 0.0001).

## Mechanical stretch and cell phenotype

To investigate the early changes in astrocytes under extra pressure load, we established an *in vitro* stretch protocol to mimic the pathological progression of glaucoma. Given that the bulk of GFAP/S100B double-positive cells were located in the ONH (Fig 3A), we dissociated the ONH and investigated GFAP/S100B double staining (Fig 3Ba).Retinal astrocytes originate from ONH [22], and our preliminary experiments showed that both retinal and ONH astrocytes are positive for GFAP and AQP9, but negative for AQP4 (S2 and S3 Figs), which is consistent with reports in the literature [23,24]. We consider the two phenotypes to be essentially the same but affected by location and surrounding tissues. Using flow cytometry to analyze cells isolated from the ONH, we found that over 95% of the cells were positive for GFAP, and cultured astrocytes showed double positive for GFAP and AQP9 (S4 Fig). This study used a consistent isolation procedure and did not differentiate between retinal and ONH astrocytes. Normal IOP is typically around 10–21 mmHg [25]. A mild increase to 25 mmHg can cause a maximum strain of 5.3% in the Lamina cribrosa of the ONH in humans (replaced by astrocytes in rodents) [26]. In our mouse model, the baseline IOP is around 8.6 ± 1.1 mmHg and can reach up to 34.6 ± 8.0 mmHg, a roughly fourfold increased. To maintain relevance to the animal experiments and observe differences within a short timeframe, primarily cultured astrocytes were subjected to continuous mechanical stretch in a parallel direction via a 4% or 20% sine wave stretch for 1 hour at 1 Hz (Fig 3Bb). The control cells were treated identically but were not exposed to stretch. Phase contrast images showed that the cells that were not exposed to stretch were essentially unchanged after 1 hour (Fig 3C and 3D). With application of 4% stretch, the astrocytes rearranged and tended to aggregate, forming cell-free patches. When 20% stretch was applied, they contracted, lost their stellate structure, and developed elongated processes (Fig 3C and 3D).

When observed using CellLight™ Actin-GFP (Fig 4), a 4% stretch was found to result in the gradual formation of ordered actin fibers (Fig 4A–4C, upper) and a slightly smaller cell shape (Fig 4D, left). After 15, 30, 45, and 60 min of stretching, the cell area was 76 ± 7%, 70 ± 11%,77 ± 7%, and 74 ± 9% of the initial area, but no observable changes from the initial polygonal cell shape were observed. When the stretch stimulus increased to 20%, actin fibers initially formed but became disrupted over time, with actin accumulation and loss of polygonal shapes (Fig 4A–4C , bottom) and cells becoming much smaller (Fig 4D, right). After a 15-, 30-, 45-, and 60min exposure to stretching, the cell area decreased to 84 ± 3%, 80 ± 9%, 73 ± 20%, and 59 ± 10% of the initial area.

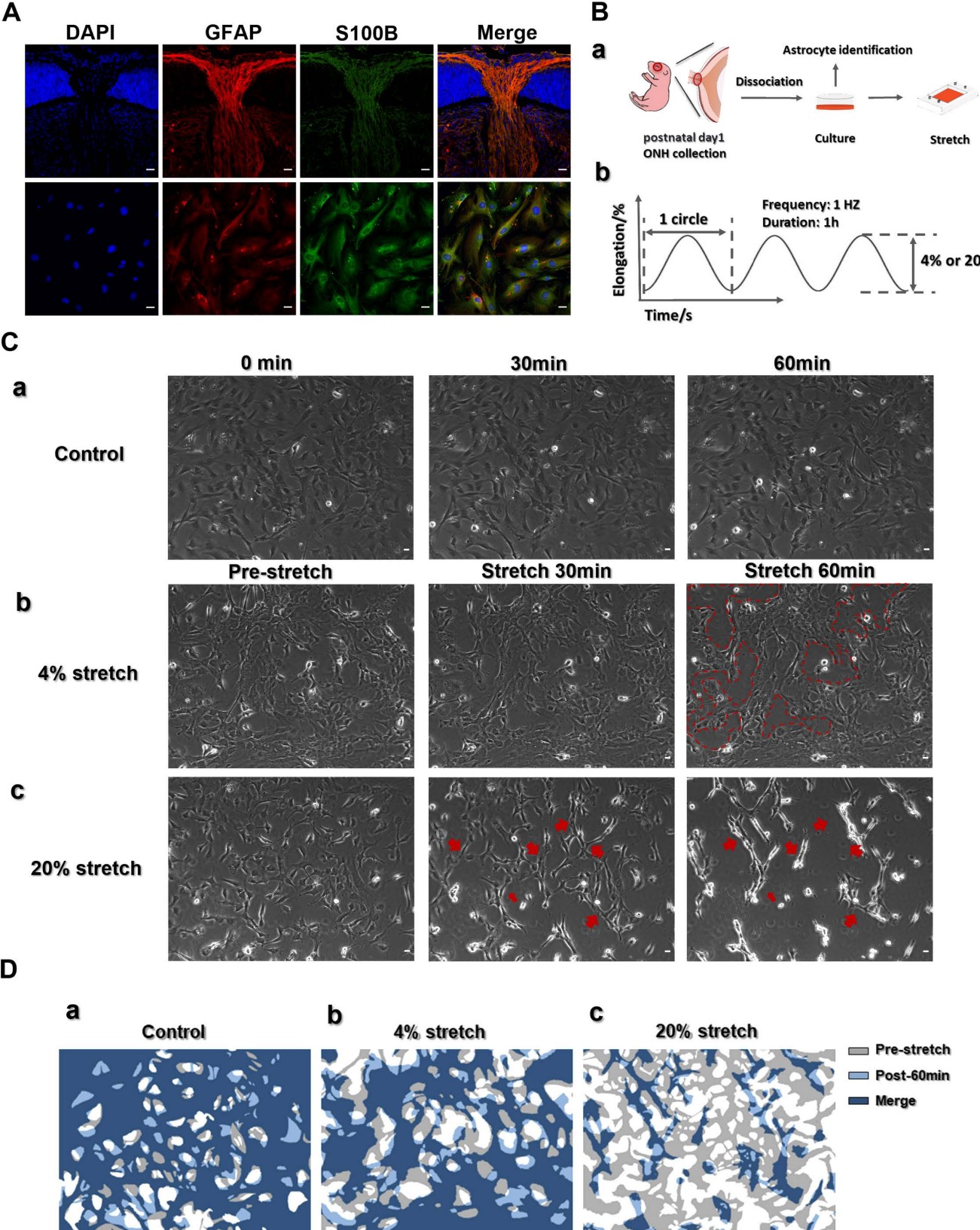

**Fig 3. Mechanical stretch and astrocyte morphology.** (A) Astrocytes from 1-day old mice in ONH sections (upper image) and shown as isolated cells (lower image) stained with GFAP (red) and S100B (green). Nuclei stained with DAPI (blue). (B) Schematic illustrating ONH astrocyte isolation/culture

procedures (a) and the stretch protocol (b). (C) The astrocytes were stretched using a sinusoidal stretch–release wave: (a) control no-stretch group; (b) 4% stretch; (c) 20% stretch. (D) Schematic of changes in cell occupancy under different stretching conditions. Gray and light blue indicate the cell-occupied area before and 1 hour after stretching. Dark blue indicates overlapping areas: (a) control group; (b) 4% stretch; (c) 20% stretch. The experiments were repeated at least three times. Scale bar: 20 μm.

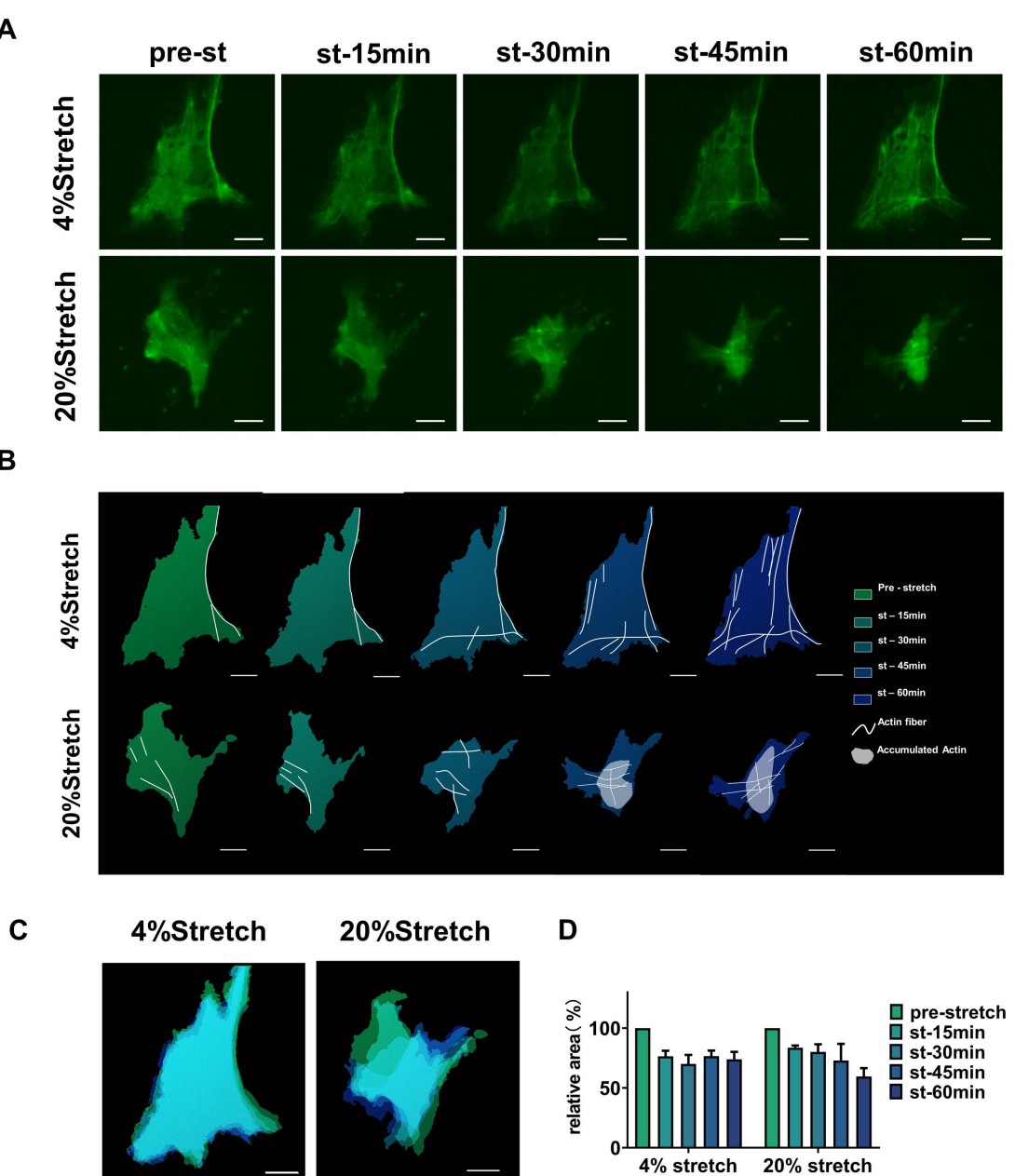

**Fig 4. Stretched ONH astrocytes display cytoskeletal reorganization.** (A) Changes in astrocyte morphology and cytoskeleton under different stretch loads, with intracellular actin visualized via infection with CellLight™ Actin-GFP, BacMam 2.0. (B) Schematic of (A). White lines represent actin fibers, white patches indicate aggregated actin protein, and the color gradient from green to blue-violet represents stretch time. (C and D) Color-coded merger of cell shapes (C) and areas (D) at each time point. ONH, optic nerve head.

## Mechanical stretch and S100B gene expression

After a further 24 h in culture after cessation of the 1-hour 20% stretch, most ONH astrocytes recovered their shape, although a small proportion did not (Fig 5A). Using Cell Count Reagent SF, we found that the stretched cells were less viable compared to controls, despite a maintained growth trend. At the 24-hour timepoint, total cell viability in the control and stretched groups was, respectively, 210 ± 21% and 163 ± 6% of their initial viability (Fig 5B). Although the viability of the stretched cells showed a decrease compared to the control cells after 24 hours of culture, there was no significant statistical difference (p = 0.1). S100B mRNA expression decreased post-stretch and was partially restored by 24 hours, with fold-changes of 1.01 ± 0.14 in controls, 0.36 ± 0.03 immediately post-stretch (p = 0.0023), and 0.69 ± 0.14 (p = 0.0541) at 24-hours post-stretch (Fig 5C). Immunostaining showed that actin in normal, unstretched cells was mostly localized on the cell membrane (Fig 5D), whereas the cells that experienced a 20% stretch for 1 hour (which were more elongated than the cells in the control group with a disturbed cytoskeleton) showed clear actin accumulation. More prominent actin fibers were detected in cells that had recovered their polygonal shape, whereas cells that recovered less well displayed no obvious increase in their actin fiber network (Fig 5E). S100B immunoreactivity remained diminished after 24 hours of culture after the end of stretch application, but mRNA expression had already begun to rebound. Collectively, these results suggest that mechanical stretching can trigger the downregulation of S100B gene expression, which in turn influences the recovery of cell morphology.

## ONH astrocytes are regulated by S100B

*In vitro* and *in vivo* experiments have shown that S100B expression decreases under mechanical load. To clarify the effects of downregulated S100B expression on ONH astrocytes, we used siRNA to silence the expression of S100B. This showed that, 24 hours after transfection, the expression of S100B in the siRNA S100B group was less than 1% of that in the siRNA control group (Fig 6A, siRNA control 1.06 ± 0.42, siRNA S100B 0.007 ± 0.00, p = 0.0002). GFAP expression in the siRNA S100B cells was approximately twice that in the control group (Fig 6B, siRNA control 1.05 ± 0.32, siRNA S100B 2.13 ± 0.86, p = 0.025). S100B-silenced astrocytes lost their polygonal shape, became smaller, slender, and elongated, with noticeable GFAP accumulation (Fig 6C–6E). This was consistent with the images of immunostained stretched cells (Fig 5D), in which actin fibers were not obviously formed or had resolved (Fig 6E).

RNA-seq also revealed distinct expression patterns in the S100B-silenced group and the control group (S5 Fig). For genes downregulated by at least five-fold (S5C Fig), the analysis showed that these genes are mainly involved in GO biological processes related to "response to nutrient", "intracellular calcium ion homeostasis", "cellular component assembly involved in morphogenesis" and in specific KEGG pathways related to "glutathione metabolism" and "motor proteins". Genes contributing to the assembly of cellular components and morphogenesis, such as Tmod1, Spink2 and Mylk3, suggest that the cytoskeleton and morphology may undergo significant changes. This aligns with our morphological observations. These genes are key to the maintenance of structural integrity, intracellular transport and adaptation to mechanical stress. Motor protein-related genes involved in actin and microtubule dynamics, including Mylpf, Dnah14, Dynlt4 and Myh7b, were also highlighted. Significantly downregulation of genes associated with calcium ion homeostasis, such as S100b, Tmem178, Slc24a2 and Myh7b, indicates a possible suppression of intracellular calcium signaling. On the other hand, genes upregulated by at least five-fold were enriched in histone acetylation and visual perception (S5D Fig). However, it is noteworthy that although qPCR analysis detected an increase in

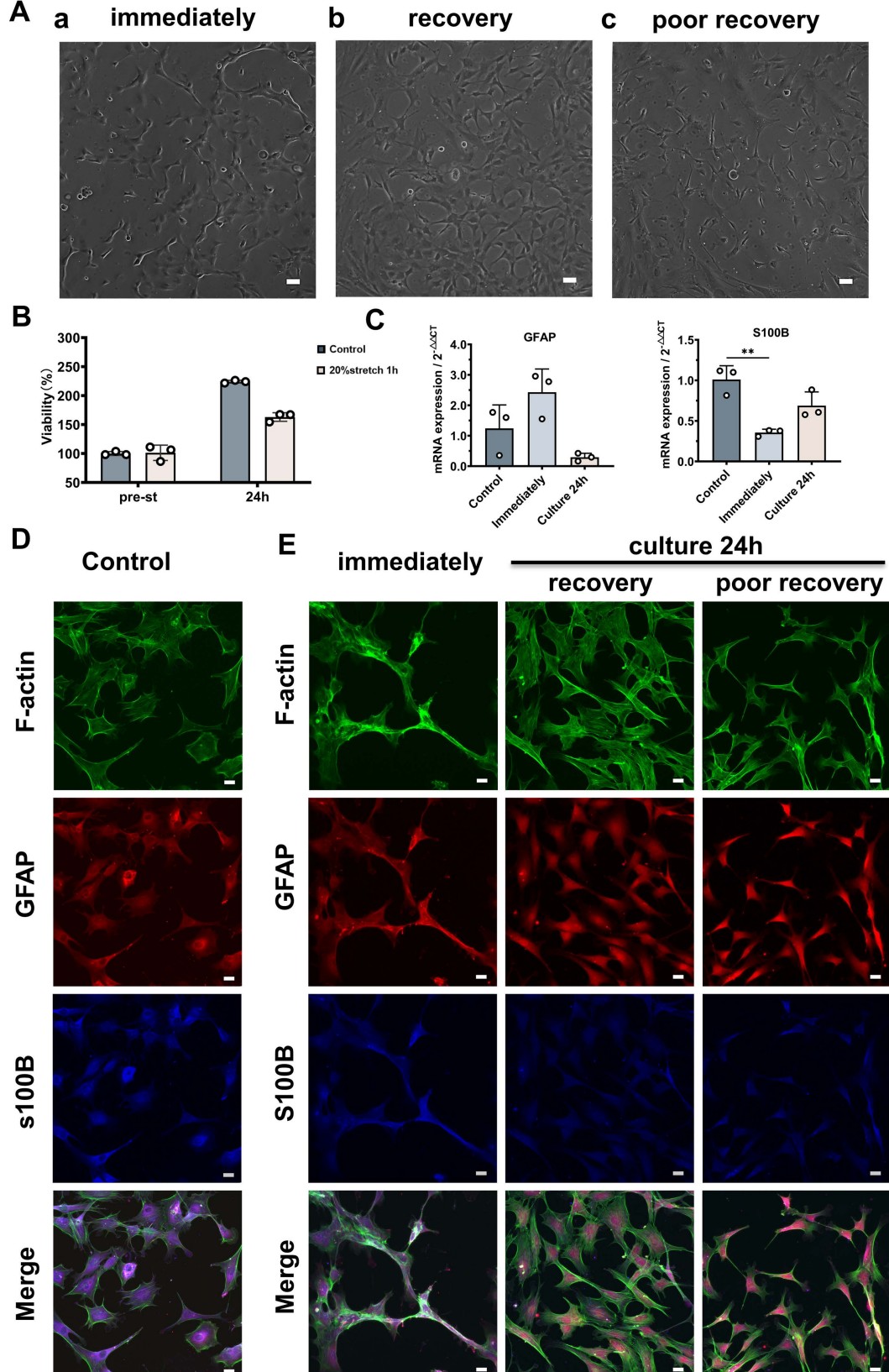

**Fig 5. Astrocyte morphology is altered by stretch-induced downregulation of S100B expression.** (A) Astrocyte cell morphology after exposure to 20%/1-Hz mechanical stretch for 1 hour. Immediately after applying the stretch stimulus, cells tended to assume a

cord-like morphology (a), but most recovered their polygonal shape after 24 hours in culture (b). On occasion, cells were small in shape and poorly extended (c). Scale bar: 100 μm. (B) Viability of the stretched cells was slightly lower than that of the control group cells. (C) GFAP and S100B mRNA expression of unstretched astrocytes and of astrocytes immediately and 24 hours after stretching. One-way ANOVA, * p < 0.05, ** p < 0.01, *** p < 0.001, and **** p < 0.0001. (D and E) Astrocytes immediately after stretching lose their polygonal shape and show actin aggregation. After 24 h of culture, although most cells recovered their shape, more stress fibers were formed and the concentration of intracellular S100B decreased. The processes of poorly recovering cells were slender, and no further formation of stress fibers was observed. F-actin (green), GFAP (red), S100B (blue). Scale bar: 20 μm. Cellular experiments were repeated 3–4 times for each group.

GFAP expression, there was considerable variability among samples, and no significant change was observed in the RNA sequencing data (S1 Viability of the stretched cells Table). This might be due to the relatively weak influence of S100B on GFAP gene expression or to the limited impact within 24 hours, given that the half-life of GFAP is much longer than S100B.

## Discussion

Astrocytes, a type of glial cell, are vital for maintaining homeostasis and responding to various environmental stimuli such as ischemia [27], inflammation [28], injury [29] or neurodegenerative disease [30]. They undergo a series of morphological, biochemical, and functional changes to adapt to changing conditions, known as "astrocyte remodeling." These changes include hypertrophy, cell proliferation, aberrant calcium signaling, and upregulation of GFAP expression, along with the release of various molecules such as cytokines, chemokines, and growth factors. Morphological changes in astrocytes are often considered one of the most characteristic features of activation.

In mice with defects in the expression of intermediate filament-associated proteins, astrocytes become smaller with shorter cell processes [31], have reduced vesicle mobility [32], are associated with autoimmune issues [33], and exhibit reduced tissue resistance to mechanical pressure [34]. Together, these factors can lead to slower scar formation [35], increased disease severity and accelerated neuron death [36]. Based on the results presented here, we argue that the weak GFAP signal in our early-stage animal model of OHT points to a possible imbalance in intermediate filament assembly–disassembly that, in turn, leads to changes in cell morphology and physical support for neurons. It is also possible that astrocyte nutrition and homeostatic capacity are suppressed or impaired, increasing neuronal vulnerability. However, the reaction was found to be transient and reversible (Figs 2D and 5A–5D), implying that the suppressed astrocytes undergo compensatory proliferation and hypertrophy in the later stages of their recovery (Fig 1).

We hypothesize that this reversible state may reflect a preconditioning state in astrocytes after exposure to mechanical stimulation. By "preconditioning state," we mean the ability of cells or tissues to better respond to subsequent stimuli and challenges after experiencing a period of moderate mechanical stimulation [37]. Preconditioning has been observed in organs such as the nervous system [38], sclera [39], heart [40], kidney [41,42], and pancreatic beta cells [43]. Known preconditioning stimulations include ischemia [44], hypoxia [44], medications (metformin [42] and calcium channel blockers [43]), physical therapy (mechanical stimulation [39] and hypothermia [45]), neurotoxic or neuroinflammatory agents (NMDA [46], 6-OHDA [47], thrombin [48], and LPS [49]), and dietary restrictions [41]. We are still unclear regarding the precise mechanism that causes astrocyte preconditioning under mechanical pressure stimulation, but, interestingly, low expression of S100B was observed in both the *in vivo* (Fig 2Eb) animal model and *in vitro* (Fig 5C) stretching experiments. Indeed, we found that S100B silencing led to slender cell morphology, the risk of increased GFAP expression, and disruption of the ordered arrangement between astrocytes (Fig 6).

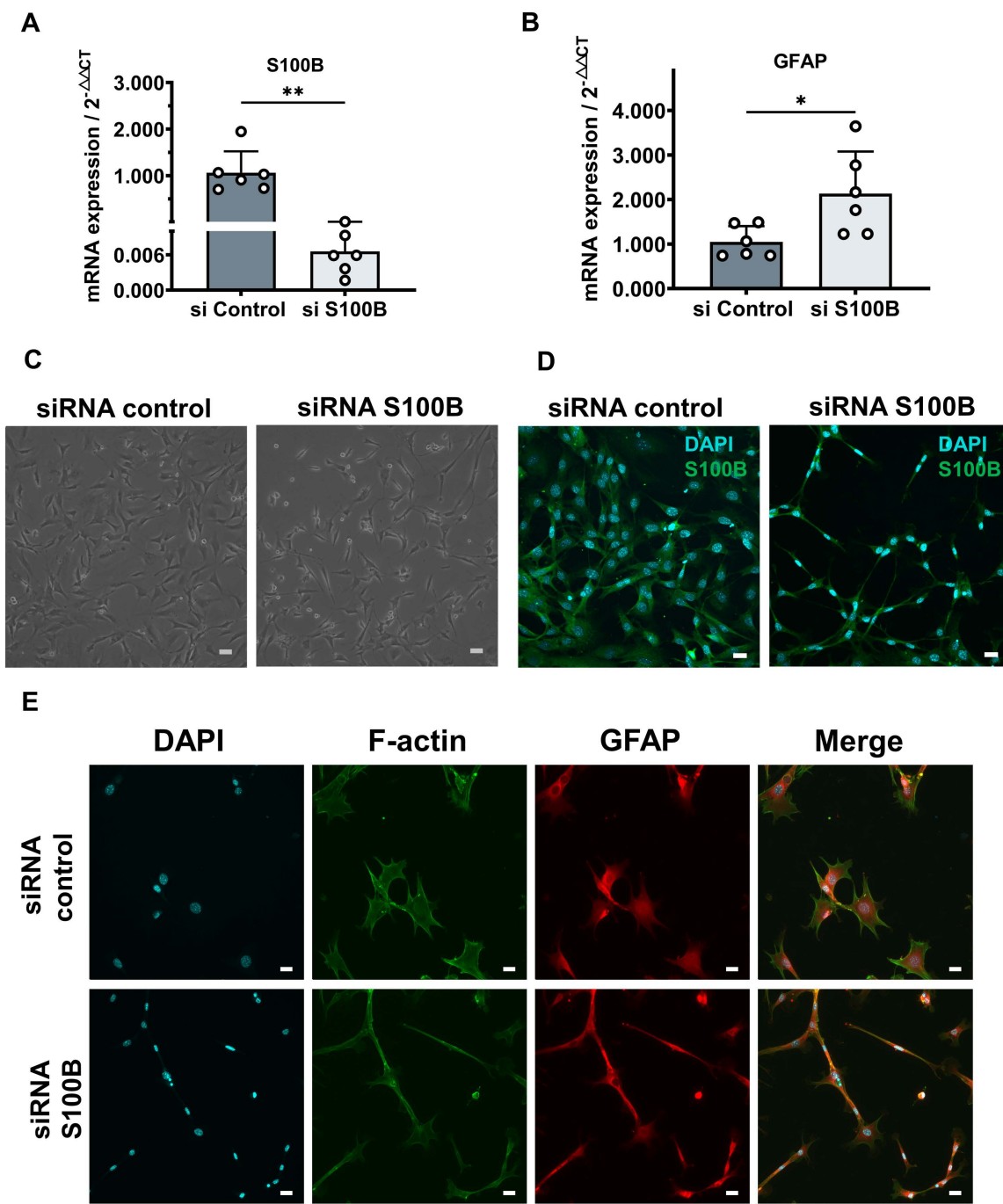

**Fig 6. S100B silencing leads to atrophy in ONH astrocytes.** (A and B) qPCR expression of S100B (A) and GFAP (B) in S100B siRNA-treated cells 24 h after transfection relative to control cells. (C) Cell morphology of the control siRNA-treated group and the S100B siRNA-treated group 24 hours post-transfection. Scale bar: 100 μm. (D) Merged images of S100B (green) and DAPI (cyan) staining of control siRNA-treated and S100B siRNA-treated cells 24 hours post-transfection. Scale bar: 20 μm. (E) Typical images of stained astrocytes with and without S100B silencing. S100B-silenced astrocytes developed linear atrophy. GFAP protein, but not actin protein, appeared to accumulate, but no significant stress fiber formation was observed. F-actin (green), GFAP (red), DAPI (cyan). Scale bar: 20 μm. Cellular experiments were repeated 3–4 times for each group.

In many neurological diseases, S100B is often considered to be a damage-associated molecular pattern (DAMPs) indicator [50] and emergency valve [14]. S100B can interfere with the polymerization of GFAP, reduce the assembly rate of GFAP filaments, and disassemble preformed glial filaments [51,52]. Moreover, it can inhibit GFAP phosphorylation activated by cAMP or $Ca^{2+}$/calmodulin [53] and induce $Ca^{2+}$-dependent microtubule disassembly, leading to the aggregation of vimentin filaments [54]. Stretch induces the formation of stress fibers or accumulation of F-actin. However, silencing S100B did not lead to significant changes in stress fibers, indicating that S100B contributes differently to the assembly of glial filaments and stress fibers. In fact, silencing S100B results in the rapid disassembly of actin fibers and enhanced formation of GFAP filaments [55]. It should be noted that the specific morphological changes vary among different cell types [55,56]. The RNA-seq data presented here also suggest that silencing may negatively impact upon cell extension and migration capabilities and calcium signaling, thereby affecting the cells' adaptability to environmental changes. Conversely, it positively influences histone acetylation, which is typically associated with increased transcriptional activity. This points to the activation of specific gene expression programs or alterations in the phases of the cell cycle in astrocytes (S5 Fig). The RNA-seq analysis not only provides strong evidence for the morphological changes observed in this study but also enhances our understanding of the underlying cellular mechanisms.

Reduced levels of S100B, which can be caused by the Borna disease virus or prenatal stress, may make neurons vulnerable [57] and affect postnatal brain development [58]. Data from human vitreous humor proteomics studies also show that S100B detection in patients with glaucoma runs at only 70% of that in individuals without glaucoma [59]; however, a full understanding of the role of S100B in glaucoma is lacking. The data presented here, however, suggest that a preconditioning state exists prior to the activation of astrocytes in our experimental models of glaucoma, and that this is associated with S100B. Preconditioning may modulate the activation potential and self-protective mechanisms of astrocytes, allowing them to better adapt to environmental stimuli and influence neural support, thereby enhancing their ability to resist disease. Although our study did not fully confirm that the transitional state we evoke is indeed a preconditioning state, there are a number of pointers, reported herein, that indicate that it is likely to be so. Accordingly, we believe that our findings are of significant importance for future explorations, offering insights for future research directions and may contribute to the development of theories and a deeper understanding of related mechanisms in the fields of astrocyte biology and glaucoma.

## Supporting information

**S1 Fig. RGC insult in OVD-induced transient OHT model.** (A) Brn3a staining of the retina. a. Retina from untreated mouse. b, c, and d. Retina from mice after OHT 4 weeks. RGCs from a different individual experienced insults of varying severity. (B) Quantification of Fluorogold fluorescence area above threshold. (C) Quantification of Brn3a fluorescence area above threshold. n = 3 mice for each group, with 12 areas imaged in each retina. Individual data points representative of individual image fields are plotted. (C) Quantitative RNA expression of Brn3a. n = 12 mice for the naïve group, n = 9 mice each for OHT post-4 and 8 weeks. Unpaired t-test, Mann–Whitney test, * p < 0.05, ** p < 0.01, and *** p < 0.001. Scale bar: 20 μm.
(TIF)

**S2 Fig. Astrocytes in murine retina wholemounts at 8 weeks.** (A) Schematic diagram of the retinal wholemount. The dashed box indicates the area observed under the microscope in (B and C). (B, upper) AQP4 was only expressed at the end-foot of Müller cells but not in

GFAP-positive retinal astrocytes. (B, lower) AQP9 (green) colocalized with GFAP-positive (red) astrocytes in the retina. Scale bar, 50 μm.
(TIF)

**S3 Fig. Astrocytes in murine ONH sections at 8 weeks.** (A–C) AQP4 was expressed in Müller cells and the optic nerve astrocytes but not in GFAP-positive ONH astrocytes. (D–F) AQP9 colocalized with GFAP in the retinal and ONH astrocytes. Scale bar, 50 μm.
(TIF)

**S4 Fig. Characterization of primary cultured ONH astrocytes.** (Aa) Immunocytochemistry of primary cultured cells for GFAP (green), which is a marker of astrocytes. (Ab) Negative control (secondary antibody only). (B and C) Flow cytometric analysis of cells. Stained cells were measured using a FACS Verse flow cytometer (BD Biosciences). Data were collected using logarithmic amplification on 10,000 cells. Alexa 488 fluorescence (X-axis) and cell counts (Y-axis) are shown. (B) The percentage of GFAP-positive cells (corrected for non-specific immunofluorescence, red line) was 72.8% in all populations. (C) The percentage of GFAP-positive cells (corrected for non-specific immunofluorescence, red line) was 95.1% in gated populations. (Da–c) No AQP4 (red) was expressed in primary cultured GFAP-positive (green) astrocytes. (Dd–f) AQP9 (red) was expressed in cell membranes and colocalized with GFAP (green). Scale bar, 20 μm.
(TIF)

**S5 Fig. Comprehensive analysis of RNA-seq data from siRNA-treated astrocytes.** (A) Distribution of gene types identified in the RNA-seq dataset. (B) Heatmap of K-means clustering of top 2,000 differentially expressed genes in control siRNA-treated and S100B siRNA-treated cells. (C) Gene Ontology (GO) enrichment analysis for genes (287 genes) with at least a five-fold downregulated expression in the S100B silenced group compared to the control group. a. Overview of enriched terms across downregulated expression genes. b. Bar graph of top-level biological processes colored by p-values. (D) Gene Ontology (GO) enrichment analysis for genes (269 genes) with at least a five-fold upregulated expression in the S100B silenced group compared to the control group. a. Overview of enriched terms across upregulated expression genes. b. Bar graph of top-level biological processes colored by p-values.
(TIF)

**S1 Table. RNA-seq analysis data.**
(XLSX)

## Author contributions

**Conceptualization:** Kenji Matsushita, Rumi Kawashima.

**Data curation:** Weiran Huang, Rumi Kawashima, Yuichi Yasukura, Kaito Yamaguchi.

**Formal analysis:** Weiran Huang, Rumi Kawashima.

**Funding acquisition:** Weiran Huang, Kenji Matsushita, Rumi Kawashima, Yuichi Yasukura, Shinichi Usui, Koichi Baba, Kohji Nishida.

**Investigation:** Weiran Huang, Rumi Kawashima, Susumu Hara, Yuichi Yasukura.

**Methodology:** Weiran Huang, Kenji Matsushita, Rumi Kawashima, Kaito Yamaguchi, Shinichi Usui.

**Project administration:** Kenji Matsushita.

**Resources:** Weiran Huang, Yuichi Yasukura, Kaito Yamaguchi, Shinichi Usui.

**Supervision:** Kenji Matsushita, Rumi Kawashima, Andrew J. Quantock, Kohji Nishida.

**Writing – original draft:** Weiran Huang, Kenji Matsushita.

**Writing – review & editing:** Kenji Matsushita, Rumi Kawashima, Yuichi Yasukura, Kaito Yamaguchi, Shinichi Usui, Koichi Baba, Andrew J. Quantock, Kohji Nishida.

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
