## [Decision Letter · Decision Letter 0]

2 Jul 2024

PONE-D-24-17014Transient ocular hypertension remodels astrocytes through S100BPLOS ONE

Dear Dr. Matsushita,

Thank you for submitting your manuscript to PLOS ONE. After careful consideration, we feel that it has merit but does not fully meet PLOS ONE’s publication criteria as it currently stands. Therefore, we invite you to submit a revised version of the manuscript that addresses the points raised during the review process.

Please, follow the suggestions and take into consideration the comments of both Reviewers. 

We look forward to receiving your revised manuscript.

Kind regards,

Silvana Allodi, PhD

Academic Editor

PLOS ONE

Journal Requirements:

3. Thank you for stating the following financial disclosure: "This study was supported by JST SPRING, Grant Number JPMJSP2138 (Next generation Researcher development program), Grants-in-Aid for Scientific Research A (No. 25253093) and C (No. 26462686) from the Ministry of Education, Culture, Sports, Science, Technology and grant number 23gm1210004 from AMED."

4. We note that your Data Availability Statement is currently as follows: "All relevant data are within the manuscript and its Supporting Information files."

5. We notice that your supplementary figures are uploaded with the file type 'Figure'. Please amend the file type to 'Supporting Information'. Please ensure that each Supporting Information file has a legend listed in the manuscript after the references list.

6. We notice that your supplementary figures are included in the manuscript file. Please remove them and upload them with the file type 'Supporting Information'. Please ensure that each Supporting Information file has a legend listed in the manuscript after the references list.

Reviewers' comments:

Reviewer's Responses to Questions

**Comments to the Author**

1. Is the manuscript technically sound, and do the data support the conclusions?

Reviewer #1: Yes

Reviewer #2: Yes

2. Has the statistical analysis been performed appropriately and rigorously? 

Reviewer #1: N/A

Reviewer #2: Yes

3. Have the authors made all data underlying the findings in their manuscript fully available?

Reviewer #1: Yes

Reviewer #2: Yes

4. Is the manuscript presented in an intelligible fashion and written in standard English?

Reviewer #1: Yes

Reviewer #2: No

5. Review Comments to the Author

Reviewer #1: In the manuscript, authors describe the effect of transient ocular hypertension and mechanical stretch on astrocytes, identifying morphological changes in these cells, and downregulation of S100B as an early response. Furthermore, similar alterations in astrocyte morphology were also observed after S100B silencing. Overall, the work is of interest, the article is well written, and methods are well described. Data is properly presented and discussed. However, some points should be addressed before publication:

Major points:

Figures 2E, 5B, 5C, 6A, 6B: for better comparison of individual values and their range, plots that include single values should be used instead of the current form of data presentation.

Line 213 / Figure 1B: The loss of FG labeling seems quite severe considering the magnitude and length of intraocular pressure increase. Is there any comparison of FG staining with immunostaining for an RGC marker? FG labeling is dependent on axonal transport and may be compromised in case of axonal disfunction. Furthermore, it is important to include a quantification of these images.

Figure 1 C-D and Figure 1 G-I: Please clarify which time point is evaluated in these experiments. 4 weeks? 8 weeks? Both?

Figure 1F and 2C: data should be represented as individual animals, and not as field images.

Figure 4. What is the sample size in this experiment? How many cells, from how many animals were analyzed? Graph 4D seem to include data from a single cell.

Line 367-369: Please include the p value for this comparison. Line 389 stated that “Viability of the stretched cells was lower than that of the control group cells”, but no statistics are shown.

To what extent does the % stretch applied in vitro correlates with the stress observed during ocular hypertension? Please discuss that.

Minor points:

Line 111: please briefly describe which criteria were used to locate the ONH.

Line 140: please provide the catalog number for all secondary antibodies.

Line 148: how are imaging fields distributed regarding center and periphery in the retina whole mounts?

Figure 1G and 1I: I suggest making the background in the curves a little more transparent (but not the lines), to make it easier to analyze and compare all the groups in the graph.

Figure 1 G-I: The word “hypertrophy” is misspelled in the legend.

Lines 274-276: The sentence related to figure Fig 1I is not very clear.

Reviewer #2: Huang et al. analyzed ONH astrocyte remodeling in a mouse model of transient ocular hypertension (OHT) and an in vitro cell stretch model. Based on morphological and gene/protein expression changes, they identified a preconditioning state in astrocytes during the early-stage of exposure to mechanical stimulation (OHT or mechanical stretch), before astrocyte remodeling occurred. Low expression of S100B seems the major feature of this preconditioning state. They further discovered that S100B silencing lead to slender cell morphology, increased expression of GFAP, and disruption of the ordered arrangement between astrocytes. Overall, these findings are very interesting regarding the mechanism of glaucoma-induced astrocyte remodeling. The author can improve the manuscript in the following aspects.

1. They described the supplementary figures in the discussion section. In general, discussion does not include figures. Please put these contents in the results section.

2. The rationale to investigate S100B in these models is not well described. As it is possible these are many genes down-regulated during the preconditioning state.

3. While the S100B silencing experiment is encouraging, we don’t know the full transcriptome changes in these cells. I suggest RNA sequencing on these cells.

6. PLOS authors have the option to publish the peer review history of their article (what does this mean? ). If published, this will include your full peer review and any attached files.

**Do you want your identity to be public for this peer review?** For information about this choice, including consent withdrawal, please see our Privacy Policy .

Reviewer #1: No

Reviewer #2: **Yes: ** Danian Chen

---

## [Author Response · Author response to Decision Letter 1]

15 Oct 2024

I would like to thank the editor and reviewers for their careful review. We have considered the reviewers’ comments carefully, performed additional experiments, and revised the manuscript. We have attached our responses to the reviewers' comments on the previous version of our manuscript below.

Reviewer #1: In the manuscript, authors describe the effect of transient ocular hypertension and mechanical stretch on astrocytes, identifying morphological changes in these cells, and downregulation of S100B as an early response. Furthermore, similar alterations in astrocyte morphology were also observed after S100B silencing. Overall, the work is of interest, the article is well written, and methods are well described. Data is properly presented and discussed. However, some points should be addressed before publication:

Major points:

1. Figures 2E, 5B, 5C, 6A, 6B: for better comparison of individual values and their range, plots that include single values should be used instead of the current form of data presentation.

Thank you for your valuable feedback. We have revised Figures 2E, 5B, 5C, 6A and 6B to include plots with individual values for better comparison of data points and their ranges, as requested.

2. Line 213 / Figure 1B: The loss of FG labeling seems quite severe considering the magnitude and length of intraocular pressure increase. Is there any comparison of FG staining with immunostaining for an RGC marker? FG labeling is dependent on axonal transport and may be compromised in case of axonal disfunction. Furthermore, it is important to include a quantification of these images.

As pointed out, evidence from FG labeling alone may not be sufficiently convincing. Thus, we have now included qPCR data and added images of Brn3a immunostaining 4 weeks after the transient increase in IOP (S1 Fig A). Images in Figure 1C depict the damage at a moderate level. In reality, at this time point, RGCs experienced insults of varying severity, from mild to severe (S1 Fig Ab-d). While we acknowledge that some RGC bodies with axonal dysfunction might still remain within the retina, the death of RGCs is definitive compared to the control group.

3. Figure 1 C-D and Figure 1 G-I: Please clarify which time point is evaluated in these experiments. 4 weeks? 8 weeks? Both?

The previous figures included data from both 4 weeks and 8 weeks, which might have caused some confusion. Thus, we have made several adjustments to the figures originally presented in the manuscript. The original Figure 1C included data from the 8-week time point; this has now been removed. The current Figure 1D now displays only the comparison between the control group and astrocytes at 4 weeks. At this time point, some astrocytes exhibit hypertrophy while others do not show this phenotype. The former Figure 1D has been split into new Figures 1E and 1F to. Figure 1E corresponds to the changes described in the new Figure 1D, while Figure 1F further contrasts hypertrophic astrocytes at different time points (4w and 8w) with control group astrocytes. We think that these modifications improve the clarity and specificity of our observations (subsequent figures in the manuscript have also been renumbered and described in line 239-324).

4. Figure 1F and 2C: data should be represented as individual animals, and not as field images.

The all figures have been revised as requested. The individual data points are now representative of individual mouse retinas, instead of images.

5. Figure 4. What is the sample size in this experiment? How many cells, from how many animals were analyzed? Graph 4D seem to include data from a single cell.

Figures 4B-4D are analyses of Figure 4A and represent the results of a single experiment. However, this experiment was repeated three times for each group. We have replaced Figure 4D with statistical data from three independent experiments. Typically, one stretching chamber (2cm x 2cm) requires astrocytes from 6-8 pup mice, and culture for 2-4 weeks.

6. Line 367-369: Please include the p value for this comparison. Line 389 stated that “Viability of the stretched cells was lower than that of the control group cells”, but no statistics are shown.

There is no significant difference between this two groups, the p-value is 0.1 (Unpaired t-test, Mann–Whitney test). We agree that the description in our previous statement was misleading, as pointed out. It has been revised to: "Although the viability of the stretched cells showed a decreasing trend compared to the control cells after 24 hours of culture, there was no significant statistical difference (p=0.1)." (Line 426-428) " Viability of the stretched cells was slightly lower than that of the control group cells " (Figure 5B legend)

7.To what extent does the % stretch applied in vitro correlates with the stress observed during ocular hypertension? Please discuss that.

Normal IOP is typically around 10-21 mmHg (PMID: 28577860). A mild increase to 25 mmHg can cause a maximum strain of 5.3% in the Lamina cribrosa of optic nerve head in humans (replaced by astrocytes in rodents) (PMID: 16249498). In our mouse model, the baseline IOP is around 8.6 ± 1.1 mmHg and can reach up to 34.6 ± 8.0 mmHg, about fourfold increased. To maintain relevance to the animal experiments and observe differences within a short timeframe, we set the maximum cell stretch load at 20%.

We have added this description to the "Result" section. (Line 376-383)

Minor points:

1. Line 111: please briefly describe which criteria were used to locate the ONH.

According to the reference, we defined the optic nerve head (ONH) as follows: "The intraocular portion of the nerve and the retrobulbar nonmyelinated portion (within a distance of 0.6–0.8 mm behind the globe)." We have added the criteria to the "Methods" section. (Line 126-128)

2. Line 140: please provide the catalog number for all secondary antibodies.

The catalog numbers for the secondary antibodies are as follows: Alexa Fluor 488 goat anti-rabbit (#A11008), Alexa Fluor 594 goat anti-rabbit (#A11012), Alexa Fluor 657 goat anti-guinea pig (ab150187). We have added this information to the "Methods" section. (Line 161-162)

3. Line 148: how are imaging fields distributed regarding center and periphery in the retina whole mounts?

To better describe the distribution of our imaging fields, we have added a schematic in Figure 1 (Figure 1B). The gray frames indicate the imaging fields.

4. Figure 1G and 1I: I suggest making the background in the curves a little more transparent (but not the lines), to make it easier to analyze and compare all the groups in the graph.

The figures have been revised as requested. (Figure 1H and 1I)

5. Figure 1 G-I: The word “hypertrophy” is misspelled in the legend.

This has been corrected. (Figure 1H and 1I)

6. Lines 274-276: The sentence related to figure Fig 1I is not very clear.

The sentence has been revised to: "In naïve eyes, S100B is mostly concentrated in the soma of astrocytes rather than the processes. However, in some transient OHT eyes, astrocytes lose this high signal of somatic S100B." (Lines 307-309)

Reviewer #2: Huang et al. analyzed ONH astrocyte remodeling in a mouse model of transient ocular hypertension (OHT) and an in vitro cell stretch model. Based on morphological and gene/protein expression changes, they identified a preconditioning state in astrocytes during the early-stage of exposure to mechanical stimulation (OHT or mechanical stretch), before astrocyte remodeling occurred. Low expression of S100B seems the major feature of this preconditioning state. They further discovered that S100B silencing lead to slender cell morphology, increased expression of GFAP, and disruption of the ordered arrangement between astrocytes. Overall, these findings are very interesting regarding the mechanism of glaucoma-induced astrocyte remodeling. The author can improve the manuscript in the following aspects.

1. They described the supplementary figures in the discussion section. In general, discussion does not include figures. Please put these contents in the results section.

Thank you for pointing this out. We have moved this content (lines 600-608) to the Results section (line 369-376).

2. The rationale to investigate S100B in these models is not well described. As it is possible these are many genes down-regulated during the preconditioning state.

The rationale for studying S100B in these models is based on its role in cytoskeletal regulation and its potential as a biomarker for neural injury. In preliminary studies, we hypothesized that astrocytes might undergo morphological changes in response to mechanical stress during the initial stages. GFAP, which forms part of the intermediate filament framework in cells and is closely associated with astrocyte activation, has been reported to be regulated by S100B. S100B, a calcium-binding protein, is primarily expressed in astrocytes within the central nervous system. It is known to be involved in various cellular processes, such as protein phosphorylation regulation, calcium homeostasis, cell proliferation, cell differentiation, and it is also one of the notable biomarkers for neural injury and repair.

Furthermore, S100B not only functions intracellularly by influencing the state of astrocytes and regulating their support and stabilization of the neurovascular unit, but can also be secreted into the extracellular space to directly affect RGCs. It has been reported that S100B can have neuroprotective effects at appropriate concentrations. Although many genes might be down-regulated during the preconditioning state, studying S100B allows us to provide a reasonable explanation for our previous findings, and to focus on a protein with diagnostic and therapeutic potential.

3. While the S100B silencing experiment is encouraging, we don’t know the full transcriptome changes in these cells. I suggest RNA sequencing on these cells.

Thank you for your valuable suggestion, we have now performed RNA sequencing on the S100B-silenced cells and the control cells to see the full transcriptome changes (S5 Fig). In this analysis, we covered a diverse array of gene types (S5 Fig A), with coding genes predominating alongside a number of pseudogenes and non-coding RNAs.

The differential expression analysis (S5 Fig B) highlights distinct expression patterns between the S100B-silenced group and the control group, while the heatmap clearly demonstrates a wide transcriptional response to siRNA treatment, including significant upregulation and downregulation of genes. These findings indicate that the modulation of S100B can indeed impact the expression of multiple genes, which is crucial for understanding the influence of S100B on gene regulation and cellular responses.

Gene Ontology (GO) enrichment analysis of significantly altered genes provides deeper insights. For genes downregulated by at least five-fold (S5 Fig C), the analysis showed that these genes are mainly involved in GO biological processes related to “response to nutrient”, “intracellular calcium ion homeostasis”, “cellular component assembly involved in morphogenesis”, and in specific KEGG pathways related to “glutathione metabolism” and “motor proteins”.

Genes contributing to the assembly of cellular components and morphogenesis, such as Tmod1, Spink2, and Mylk3, suggest that the cytoskeleton and morphology may undergo significant changes. This aligns with our morphological observations. These genes are key to the maintenance of structural integrity, intracellular transport, and adaptation to mechanical stress. Motor protein-related genes involved in actin and microtubule dynamics, including Mylpf, Dnah14, Dynlt4, and Myh7b, were also highlighted. Moreover, the significant downregulation of genes associated with calcium ion homeostasis such as S100b, Tmem178, Slc24a2, and Myh7b, indicates a possible suppression of intracellular calcium signaling. These findings suggest that silencing S100B could negatively impact cell extension, migration capabilities, and calcium signaling, thereby affecting the cells' adaptability to environmental changes.

On the other hand, genes upregulated by at least five-fold were enriched in histone acetylation (involving genes H3c3, H3c6, H4c3, H4c12, and H2bc9) and visual perception (S5 Fig D). Histone modifications are typically linked with transcriptional activity, suggesting that specific gene expression programs may be activated or that there may be alterations in the phases of the cell cycle in astrocytes.

Overall, this new RNA sequencing analysis not only provides strong evidence for the morphological changes observed in this study but also enhances our understanding of the underlying cellular mechanisms. Furthermore, it may offer valuable insights for future research aimed at targeting these pathways for therapeutic interventions. We have added this description to the "Result" (Line 471-491). and "Discussion" sections (Line554 -561).

---

## [Decision Letter · Decision Letter 1]

28 Oct 2024

Transient ocular hypertension remodels astrocytes through S100B

PONE-D-24-17014R1

Dear Dr. Matsushita,

We’re pleased to inform you that your manuscript has been judged scientifically suitable for publication and will be formally accepted for publication once it meets all outstanding technical requirements.

Kind regards,

Mohd Akbar Bhat

Academic Editor

PLOS ONE

Additional Editor Comments (optional):

Reviewers' comments:

Reviewer's Responses to Questions

**Comments to the Author**

1. If the authors have adequately addressed your comments raised in a previous round of review and you feel that this manuscript is now acceptable for publication, you may indicate that here to bypass the “Comments to the Author” section, enter your conflict of interest statement in the “Confidential to Editor” section, and submit your "Accept" recommendation.

Reviewer #1: All comments have been addressed

Reviewer #2: All comments have been addressed

Reviewer #3: All comments have been addressed

2. Is the manuscript technically sound, and do the data support the conclusions?

Reviewer #1: Yes

Reviewer #2: Yes

Reviewer #3: Yes

3. Has the statistical analysis been performed appropriately and rigorously? 

Reviewer #1: Yes

Reviewer #2: Yes

Reviewer #3: Yes

4. Have the authors made all data underlying the findings in their manuscript fully available?

Reviewer #1: Yes

Reviewer #2: Yes

Reviewer #3: Yes

5. Is the manuscript presented in an intelligible fashion and written in standard English?

Reviewer #1: Yes

Reviewer #2: Yes

Reviewer #3: Yes

6. Review Comments to the Author

Reviewer #1: I consider that the authors have now sufficiently addressed all the comments.

I have only a few minor comments:

-Lines 251-253: please take a look at the units used to describe the results of labeling area. Both μm2 and a.u. are used, while the graphs (Figure S1) show μm2.

-In Figure 3C, the labeling of each channel in the image is not described.

Reviewer #2: This rebuttal addressed all my concerns, especially RNA sequencing generated sufficient evidence to support their conclusion. Thus, I suggest accepting this work now.

Reviewer #3: The authors have incorporated all the suggestions recommended by the reviewers.

The manuscript can be accepted for publication

7. PLOS authors have the option to publish the peer review history of their article (what does this mean? ). If published, this will include your full peer review and any attached files.

**Do you want your identity to be public for this peer review?** For information about this choice, including consent withdrawal, please see our Privacy Policy .

Reviewer #1: No

Reviewer #2: No

Reviewer #3: No

---

## [Editor Report · Acceptance letter]

PONE-D-24-17014R1

PLOS ONE

Dear Dr. Matsushita,

I'm pleased to inform you that your manuscript has been deemed suitable for publication in PLOS ONE. Congratulations! Your manuscript is now being handed over to our production team.

Kind regards,

on behalf of

Dr. Mohd Akbar Bhat

Academic Editor

PLOS ONE